# The Interplay between Ghrelin and Microglia in Neuroinflammation: Implications for Obesity and Neurodegenerative Diseases

**DOI:** 10.3390/ijms232113432

**Published:** 2022-11-03

**Authors:** Cristina Russo, Maria Stella Valle, Antonella Russo, Lucia Malaguarnera

**Affiliations:** 1Section of Pathology, Department of Biomedical and Biotechnological Sciences, School of Medicine, University of Catania, 95123 Catania, Italy; 2Laboratory of Neuro-Biomechanics, Department of Biomedical and Biotechnological Sciences, School of Medicine, University of Catania, 95123 Catania, Italy; 3Section of Physiology, Department of Biomedical and Biotechnological Sciences, University of Catania, 95123 Catania, Italy

**Keywords:** Ghrelin, Ghrelin receptor, microglia, inflammation, neuroinflammation, neurodegenerative diseases, Alzheimer’s disease, Parkinson’s disease

## Abstract

Numerous studies have shown that microglia are capable of producing a wide range of chemokines to promote inflammatory processes within the central nervous system (CNS). These cells share many phenotypical and functional characteristics with macrophages, suggesting that microglia participate in innate immune responses in the brain. Neuroinflammation induces neurometabolic alterations and increases in energy consumption. Microglia may constitute an important therapeutic target in neuroinflammation. Recent research has attempted to clarify the role of Ghre signaling in microglia on the regulation of energy balance, obesity, neuroinflammation and the occurrence of neurodegenerative diseases. These studies strongly suggest that Ghre modulates microglia activity and thus affects the pathophysiology of neurodegenerative diseases. This review aims to summarize what is known from the current literature on the way in which Ghre modulates microglial activity during neuroinflammation and their impact on neurometabolic alterations in neurodegenerative diseases. Understanding the role of Ghre in microglial activation/inhibition regulation could provide promising strategies for downregulating neuroinflammation and consequently for diminishing negative neurological outcomes.

## 1. Introduction

Over the last few years, our knowledge of Ghrelin (Ghre) has increased significantly. In fact, the peptide Ghre is involved in several cellular activities affecting the gastrointestinal and immune systems. This orexigenic hormone not only regulates food intake and energy content but also modulates plasticity and cognition in the central nervous system (CNS). Ghre signaling deregulation is involved in the pathophysiology of obesity and may provide a link between metabolic syndromes and cognitive impairment [1]. Indeed, in obesity, impaired metabolic homeostasis is frequently associated with the severity of age-related cognitive decline, hippocampal function and neurodegenerative diseases [2]. These alterations are associated with increased microglia activation and synaptic profiles within microglia and lesser dendritic spines [2]. In fact, it has been reported that pharmacological inhibition of the phagocytic activity of microglia is sufficient to avoid cognitive deterioration [2].

During inflammation, proinflammatory cytokines and immune-derived cells stimulate hormone release and metabolism. Several reports show that Ghre and its receptor play a regulatory role during inflammation [3]. In the CNS, the main sources of proinflammatory molecules are glial cells, classifiable into two, large groups: microglia and macroglia. Microglial cells derive from the embryonic mesoderm. They protect the brain from injuries and diseases. Macroglia cells derive from the neuroectoderm and provide trophic support, maintain metabolism and homeostasis, and produce the myelin sheath around axons [4].

Both Ghre and microglia are involved in the pathophysiology of neurodegenerative diseases characterized by neuronal damage such as Alzheimer’s disease (AD) and Parkinson’s disease (PD) [5,6].

The focus of this review is on the mechanisms by which Ghre modulates microglia activity during obesity-induced neuroinflammation by emphasizing the effects of Ghre in inducing these cells towards an anti-inflammatory phenotype, and then how these mechanisms impact neural plasticity and cognition. The understanding of this peptide’s functions will allow for the development and implementation of new therapeutic and neurological diagnostic strategies.

## 2. Expression and Functions of Ghre and Ghre Receptor

Ghre is a small peptide of 28 amino acids, which is involved in several physiological functions [7]. Originally described as an endogenous growth hormone receptor ligand, Ghre was found to be largely produced by a population of neuroendocrine cells, rodent X/A-like cells and human P/D1 cells, found in the oxyntic mucosa of the stomach fundus [7]. Successively, Ghre gene expression was found in the hypothalamus, cerebral cortex, brainstem, heart, lung and testis [8], In addition, it is produced by neurons, glial cells, immune cells and endocrine cells located in other peripheral tissues, such as the intestines and pancreas [9]. In a sequence of catalytic steps, the precursor pre-proGhre is expressed, cleaved to pro-Ghre and transported to the Golgi body, where it is acylated by the action of the Ghre-O-acyltransferase enzyme, and undergoes a post-translational modification at serine residue 3 [10]. In the end, following translocation to the endoplasmic reticulum, pro-Ghre is further processed by prohormone convertase 1/3 to generate the anorexigenic hormone Ghre. Therefore, once secreted, it is released into the bloodstream in two distinct molecular structures: desacyl-Ghre and its acylated form [11]. This structural modification is an important step because through the circulatory system, Acyl-Ghre is able to cross the blood–brain barrier (BBB), carrying out its functions at the brain level on hypothalamic nuclei, portions of the cortex, amygdala, hippocampus and dorsal vagal complex [1,12,13] (Figure 1). In the acylated form, Ghre fulfills a wide range of physiological functions such as regulation of food intake, gastrointestinal motility and acid secretion, cardiac function, osteoblast proliferation, bone maturation and muscular/myoblast outgrowth, the formation of long-term memory, sleep cycle, the control of behaviors such as spontaneity, anxiety, and food/reward behavior, as well as the modulation of the circadian rhythm [11]. In addition, the metabolic hormone Acyl-Ghre leads the secretion of growth hormone by the pituitary gland, reduces insulin, increases glucagon secretion by pancreatic cells and promotes the hepatic release of glucose into the blood, retaining steady plasma glucose levels during fasting [14]. Furthermore, Acyl-Ghre induces the orexigenic peptide neuropeptide Y (NPY) expression and agouti-related protein (AgRP) in the hypothalamus to stimulate appetite [11].

Acyl-Ghre exerts these functions, acting through its related G-protein-coupled receptor (GPCR), known as the growth hormone secretagogue receptor (GHSR). GHSR exists in two isoforms: GHSR-1A and its truncated and nonfunctional splicing variant GHSR-1B [15]. Only GHSR-1A is able to interact with Acyl-Ghre. GHSR-1A is broadly transcribed in various crucial regions of the brain, such as the hippocampus, hypothalamus, cortex, ventral tegmental area, spinal cord, dorsal and median raphe nuclei, sympathetic preganglionic nerves and endothelial cells of the cerebral vasculature. Further, it is expressed by several immune cells and in peripheral tissue [16]. In addition, Ghre and its receptor have been found to be expressed in B and T cells, monocytes and natural killer cells [17]. In CNS, Ghre and GHSR-1A act as neuropeptides in neural transmission and function. They have been identified in the olfactory bulb where they process olfactory signals [18,19]. GHSR-1A is induced by Ghre binding and triggers its signaling pathway [20]. In contrast, The GHSR-1B isoform is unable to bind Ghre but it modulates the GHSR-1A receptor through heterodimerization processes [21]. GHSR-1B causes a conformational restriction of the GHSR-1A receptor, inactivating it and exerting a negative effect [20,22]. GHSR-1A is able to interact with GHSR-1B and other G-coupled receptors, such as serotonin, dopamine (D1 and D2) and melanocortin [23].

## 3. Role of Ghre in Neuroinflammation and Neurometabolism

An unhealthy diet can generate obesity and altered metabolism, which induces a chronic proinflammatory metabolic phenotype (metaflammation) and associated brain damage [24]. The brain inflammation due to “*wrong food habits*” confirms the existence of the gut–brain axis. Metaflammation is produced by the dysfunction of the immune metabolism. The nutrition burden triggers signaling pathways and cascades without severe immune response symptoms, but is comparable to a chronic immune response for a long time [25]. The metaflammation can also damage the microstructure of brain regions and cause abnormal physiology. Therefore, pathological obesity leads not only to weight gain but also to disorders in the human immune system and neuroinflammation [26]. Bioenergetic changes and oxidative stress stimulate microglia-driven neuroinflammation. It is well known that the CNS is immune-privileged, largely protected from the circulating inflammatory pathways. Nevertheless, altered immune responses and proinflammatory mediators impair the BBB. Tight junctions and basal lamina due to loss of control in the production of matrix metalloproteinases (MMPs) and their inhibitors tissue inhibitors of metalloproteinases (TIMPs) causes structural and functional integrity of the BBB, which in turn promotes a massive migration of leukocytes through the BBB [27]. This process contributes to triggering the activation of microglia and the neuroinflammatory response. During brain impairment, BBB disruption can occur. Some studies have shown the ability of desacylated Ghre to enhance BBB integrity [28]. Ghre is involved not only in the regulation of energy intake but also in the adjustment of the immune response by inflammatory factors [29]. As previously reported, Ghre affects the physiologic processes of several systems acting on specialized innate cells, including glial cells, macrophages and dendritic cells [30]. Notably, Ghre significantly alleviates excessive inflammation and reduces damage to different target organs mainly by reducing the secretion of inflammatory cytokines, including interleukin-6 (IL-6), interleukin-1β (IL-1β), and tumor necrosis factor-α (TNF-α) [3,31]. Proinflammatory cytokine release and Ghre induction are related to transcription factor nuclear factor kappa-B (NF-κB) expression reduction and IL-8 secretion [32]. Moreover, it is well known how inflammasomes play a key role in the development and progression of diseases. Studies have shown that Ghre inhibits the NLR family pyrin domain containing 3 (NLRP3) inflammasome activation and IL-1β maturation [33]. In microglia, Ghre promotes inactivation, inhibiting the expression of TNFα, IL-1 and NOS [30]. The Ghre inhibitory effect also occurs in downstream proinflammatory cytokines such as the transcription factor high-mobility group protein B1 (HMGB1). In both humans and animals, during severe sepsis, it is possible to find high levels of systemic HMGB1. Chorny et al. demonstrated that Ghre administration reduced HMGB1 circulating levels, removing lethality [34].

## 4. Ghre and Obesity

Ghre, being involved in the processes of immune metabolism and inflammaging, is also implicated in eating disorders and obesity. In particular, obesity is a condition that results from a chronic disruption of energy balance related to the accumulation of body fat. It is a very widespread health problem with a multifactorial etiology that includes genetic, metabolic and lifestyle factors [35]. The mechanisms underlying the reduced Ghre levels in obese patients are unclear. At the hypothalamus level, Ghre, together with insulin and leptin, can desensitize their receptors [36]. In a physiological state, Ghre levels increase due to the low availability of nutrients, stimulating their acquisition through the development of feeding behavior. Following nutrient gain, Ghre levels decrease to physiological values [37]. Ghre is also involved in lipogenesis because it stimulates the accumulation of intracytoplasmic lipids and promotes adipose tissue increase [38]. Moreover, it is able to reduce glucose-dependent insulin secretion, leading to blood glucose increase and glucose tolerance impairment [39,40]. Plasma lipid levels are able to modulate Ghre. Indeed, FFA load inhibits the GHSR effects of Ghre via the modulation of hypothalamic exogenous somatostatin release [41]. Additionally, sustained stimulation with unsaturated fatty acids triggers Ghre, possibly through an enhancement in lipid GHSR expression [42].

Therefore, unsaturated or saturated plasma lipids trigger microglia, resulting in positive or negative Ghre activation, respectively. Nevertheless, it is still unclear if overnutrition during maternal programming initiates hypothalamic Ghre signaling in offspring, hence stimulating food intake in adulthood [43]. Obesity generates Ghre resistance through an impaired response of the hypothalamic NPY/AgRP pathway to Ghre plasma concentrations. Furthermore, the neuroendocrine axis is also altered to control food intake [44]. In obese subjects, fasting plasma Ghre levels are decreased, as well as fasting insulin levels [45]. As regards the genetic aspect of human obesity, some polymorphisms of Ghre have been found related to body mass index (BMI) variation, but the specific genes mediating susceptibility are not yet known [46]. Additional pathways that take place in obesity are the Ghre/AMPK/mTOR pathways. It must be said that obesity decreases autophagy via the mechanistic target of rapamycin complex 1 (mTORC1) activation and reduces the function of AMP-activated kinase (AMPK) and sirtuin-1 (SIRT1). Crosstalk between nutrient-sensing pathways that converge on mTORC1, AMPK and SIRT1 is mediated by changes in NAD+ levels. Autophagy exerts a double role in neurodegenerative diseases. Moderate autophagy eliminates dead cells preserving the axon’s homeostasis. In contrast, disproportionate autophagy induces autophagic cell death [47]. Altered autophagy activity in the neural system has been observed in neurodegenerative diseases such as AD and PD [48]. For instance, in an in vitro model of AD, Ghre improved cell viability and increased GHSR-1A expression in a time-dependent manner in the cell line derived from the SK-N-SH neuroblastoma cell line (SH-SY5Y). Recently, it has been reported promoting autophagy was involved in Ghre against MPTP-induced neurotoxicity [49]. The regulation of autophagy may be associated with the neuroprotective effects of Ghre in neural system diseases. Therefore, regular autophagy activity can be considered as an attractive therapeutic target for neurological diseases [50]. Ghre stimulates autophagy through PI3K/AKT/MTOR inhibition and ERK1/2-MAPK activation in cortical neurons [51]. In contrast, it has been demonstrated that Ghre protected adult rat hippocampal neural stem cells from disproportionate autophagy during oxygen-glucose deprivation (OGD). Ghre treatment suppresses OGD-induced excessive autophagy activity, evidenced by a reduced expression of Beclin-1 and LC3-II and enhanced expression of p62 [52]. Furthermore, Ghre regulates the proteolytic pathways and exerts an important role in the interaction between the ubiquitin-proteasome system (UPS) and autophagy [53]. AMPK chronic activity leads to the promotion of hyperphagia and obesity, while mTOR inhibition decreases the Ghre orexigenic function [54]. Ghre can regulate autophagy improving lifespan and cognitive functions in neurons [55]. Further studies are necessary to better understand the mechanisms behind this phenomenon. Moreover, during an out-of-food period, Ghre resistance protects against weight gain and optimizes the energy status. According to this, Ghre inhibition activity may play a role in avoiding the pendulum body weight condition that occurs when following an incorrect dietary restriction [56].

## 5. Microglia: Friends and Foes of the CNS

It is well known that, in the CNS, microglia play an important role in immune surveillance.

Microglia have oval-shaped nuclei and slender, elongated processes that help them to move through chemotaxis [57]. They are equipped with a great number of receptors that act as sensors of the brain environment, allowing the monitoring role of these cells. In the course of inflammation or after cell death, specific proinflammatory or necrosis signals activate the surrounding microglia, which in turn release important molecules to avoid additional damage to the neuronal network. Indeed, these cells are uniformly distributed, and each of them occupies a well-defined surface that never overlaps that of another microglial cell. This microglia function is based on the use of an innate program providing swift control of invading pathogens and setting the stage for the arrival of adaptive immune cells such as T and B cells. In the immune response, microglial cells act in different ways releasing proinflammatory molecules by which immunological cells could intervene acting as antigen-presenting cells. Moreover, they activate CD4+ and CD8+ T-lymphocytes, exposing major histocompatibility complexes (MHC)-I and II. The release of signals by microglia is necessary to support the communication between neurons and astrocytes for the repair and reorganization of impaired synapses. Additionally, microglia are involved in the elimination of cellular debris and dead neurons. Through phagocytosis, they digest infectious agents, neurofibrillary plaques, apoptotic or necrotic cells. Dying neurons recruit microglia through the release of factors that activate phagocytosis. This is also important to maintain healthy neural networks. Triggering receptors expressed on myeloid cells 2 (TREM2) signaling, on the one hand, decreases the proinflammatory potential of microglia proinflammatory cytokines such as IL-1β, TNF-α, and NOS2, while on the other, it promotes chemokine receptor 7 expression and chemotaxis. Moreover, injured neurons promote the phagocytosis of apoptotic cells through the release of milk fat globule–epidermal growth factor 8 (MFG-E8) and the activation of CD47 [58].

Microglia have phagocytic activity by promoting the release of proinflammatory cytokines and acting to remove damaged neurons. Similar to macrophages, microglia can polarize and can be activated in different ways showing two different phenotypes: cytotoxic M1, proinflammatory, stimulated by the phenomena of neuroinflammation; and cytoprotective M2.

After the stimulation of the toll-like receptors (TLR), M1 triggers the immune response and releases proinflammatory cytokines such as TNF-α, IFN-y, IL-1, IL-6 and IL-12. These cytokines increase oxidative stress through ROS production, upregulation of iNOS and nitrogen free radicals, to trigger apoptotic mechanisms [59]. M1 activation leads to proinflammatory mediator release and promotes neuroinflammation, which is the major pathological feature of neurodegenerative diseases.

M2 promotes the release of IL-4, IL-10 and growth factors to recover injured tissue and for regeneration. M2 has neuroprotective functions such as the inhibition of inflammation and the restoration of homeostasis [60,61]. Nevertheless, this classification is based on a simplified classical binary identification of microglia. Microglia show overlapping functional conditions shifting from one to the other, mainly influenced by the activated pathways. Some microglia belong to an intermediate phenotype since they may produce both proinflammatory and anti-inflammatory markers [62,63]. For instance, M2a-like microglia stimulated with IL-4 and IL-13 induce arginase, chitinase 3-like 3 and CD206. However, the M2b phenotype, since it secretes both inflammatory and restorative markers, constitutes an intermediate state of microglia. Moreover, following the resolution of the inflammatory state, M2c induces tissue remodeling and restoration, producing the transforming growth factor beta (TGF-β), CD206 and CD163 [62,64]. All the mediators produced by polarized microglial cells contribute to pain perception activating nociceptive neurons in the CNS, suggesting that reactive microglia are involved in the pathological processes of the nervous system.

## 6. Ghre, Microglia and Food Intake

Ghre has not been detected in microglia. Nevertheless, Ghre, as an anti-inflammatory hormone, inhibits microglia activation and reduces the percentage of M1 microglia [65]. Understanding the dynamics existing between microglia and the Ghre peptide is necessary. Indeed, microglia are fundamental in the regulation of feeding behavior, and their ablation leads to a decrease in food intake.

It has been reported that microglia ablation is strongly related to an increase in Ghre levels consistent with negative energy balance [66]. In fact, microglia are sensitive to nutritional variations. In vivo experiments found that a hypothalamic microgliosis associated with high proinflammatory gene expression occurs following a high-fat diet [67]. Hypothalamic inflammation is strongly involved in the development of diet-induced obesity and in systemic metabolic defects. Hypothalamic microglia are involved in the early phase of the inflammatory response [68]. Immediately after a high-fat diet (HFD), free fatty acids (FFAs) increase in the arcuate nucleus of the hypothalamus, activating microglia [69]. Hence, they change morphologically and initiate transcriptional activity, which promotes chemokine, cytokine, ROS and NO production in the hypothalamic microenvironment [70]. A long-lasting fat-rich diet and the persistent increase in FFAs and inflammatory cytokines in neural tissue cause the release of chemokines, including monocyte chemoattractant protein-1 (MCP-1) and chemokine ligand 1 (CX3CL1) [71]. These chemokines induce the migration of peripheral monocyte-derived macrophages such as CD169+ and CCR2+ from the blood and cerebrospinal fluid (CSF) into the hypothalamus [70]. The major hypothalamic areas involved in the regulation of food intake and body weight are arcuate (ARH), ventromedial (VMH), dorsomedial (DMH) and lateral (LH) nuclei. During the regulation of food intake, ARH neurons release, on the one hand, peptides able to enhance the food-intake neuropeptide Y (NPY) and agouti-related peptide (AgRP), both with orexigenic properties. On the other hand, peptides with anorexigenic properties, known as proopiomelanocortin (POMC) and cocaine amphetamine-related transcript (CART), are able to reduce food intake. Ghre stimulates food intake through NPY and AgRP in the ARH nucleus [23]. Moreover, it has been found in VMH, DMH, PVN and LH hypothalamic nuclei where Ghre increases the orexigenic signal of orexin coordinating gastrointestinal functions [72,73]. Following a caloric diet, microglia activation occurs, establishing Ghre resistance [74]. On the contrary, genetic ablation of microglia promotes anorexia and weight loss [75]. Overall, these data indicate Ghre action on microglia, but the discrepancy, however, lies in the fact that these cells do not express the Ghre receptor.

## 7. Obesity and Neurodegenerative Diseases

Much evidence suggests that nutritional variations and obesity can influence cognitive impairment and the development of neurodegenerative diseases [76]. Structural changes such as decreased hippocampal volume and atrophy have been observed in the brains of obese subjects [77]. Inflammation of the hypothalamus due to a high-fat diet induces the nuclear-factor-dependent inflammation inhibitor kappa-B (IKKβ)/NF-kB and alters the feeling of fullness by promoting susceptibility to obesity [78]. An inverse relationship among BMI, diabetes, brain volume, neuronal viability and gliosis in the hypothalamus has also been detected [79]. Similarly, obese individuals with a higher BMI but no cognitive defects showed decreased gray matter and neurodegeneration with cerebral atrophy in the frontal lobe, hippocampus and thalamus, when compared to nonobese subjects [80]. In addition, obesity alters insulin metabolism and the signaling pathway. Furthermore, obesity impairs the glucose transport mechanisms in the brain by promoting vascular deficiencies, leading to cognitive decline [81].

Depending on the type of neurons and the brain area affected, neurodegenerative diseases can have different courses, leading to neuronal structure and function alteration and causing neuron death. These debilitating disorders involve several triggering factors and, in some cases, the mutation of a specific gene that causes the mutated protein expression to modify neuronal function, promoting degeneration and neuronal death. All these progressive losses occur in pathologies such as AD or PD. Both diseases are identified as proteinopathies; in fact, the presence of misfolded protein is one of their common characteristics. The peculiarity of these altered proteins resides in their ability to build aggregates accumulating within and between the neurons, forming amyloid plates and Lewy bodies [82]. AD develops the deposition of Aβ plaques and neurofibrillary tangles, while in PD, DA neuron impairment and Lewy bodies, which are mostly composed of α-synuclein, occur [83,84]. Depending on the types of neurons and the brain area affected, neurodegenerative diseases can have different courses. It is interesting that both disorders present hyposmia as preclinical signs. In fact, even before cognitive and motor impairment, olfactory system dysfunction is the first alarm bell [85]. Remarkably, olfactory learning and memory occur under Ghre control mechanisms influencing different brain circuits (Figure 2).

The role of inflammation in neurodegenerative diseases has been highlighted in various experimental and clinical contexts. Caloric restriction is strictly related to the reduction of inflammation because it decreases proinflammatory cytokine biosynthesis.

## 8. Microglia, Ghre and Neurodegenerative Diseases

Microglia are important in neurogenesis and synaptic modification and in the reorganization of synaptic networks [86]. It has recently been reported that following an activating stimulus, they acquire a pseudo-rest state or a triggered state, which makes them more perceptive to subsequent stimuli, which promotes an exacerbated response [87]. Metabolism and energy status variations impact significantly on immune system cells. These particular cells are sensitive to circulating hormones and nutrient variations influencing immune responses and cytokine expression [88]. Microglia are significantly activated by obesity or high-fat-diet–induced cognitive impairment. After neuroinflammation in neurodegenerative diseases, microglia and astrocytes increase the central release of proinflammatory cytokines. Therefore, the new triggered microglia stimulate severe neuroinflammation, which can cause neurodegeneration and neurological disorders. Studies conducted on dopaminergic (DA) neurons have delineated how Ghre’s anti-inflammatory effects may result from the attenuation of the release of inflammatory cytokines, such as IL-6 and the increase in the anti-inflammatory cytokine IL-10, suggesting both an anti-inflammatory and a neuroprotective role [89]. It has been shown that Ghre decreased p38MAPK and pro-NGF levels and inhibited ROS production, attenuating c-jun N-terminal kinase (JNK) activation in LPS-stimulated microglia cells. In addition, a significant proinflammatory response with increased NLPR3 inflammasome was observed in the high-fat-treated BV2 cell line, suggesting that microglia are activated in obesity and that NLRP3 is involved in the regulatory mechanism of microglia-induced inflammation under high-fat diet conditions. The NLRP3 inflammasome is a multiprotein complex that is involved in the initiation and development of many diseases, such as metabolic syndrome (MetS) and type 2 diabetes [90,91]. In pathological conditions, in response to pathogen-associated molecular patterns (PAMPs) or danger-associated molecular patterns (DAMPs), the recognition of TLRs induces NLRP3 in the cytoplasm. Particularly, TLR4 participates in the initiation of neuroinflammation in various neurological diseases. TLR4 regulates NLRP3 inflammasome activation and microglial pyroptosis. Therefore, there exists an intricate association between TLR4 and NLRP3 in the regulation of microglial activation and in providing a potential neuroprotective or neurodegenerative effect in neurological disease [67,92]. In both in vivo and in vitro studies, Ghre has been shown to stimulate both astrocytes and glial cells to release cytokines acting as anti-inflammatory mediators [93]. Ghre administration downregulates TLR4 expression and then inhibits the NLRP3 inflammasome [94]. Excessive ROS contribute to the inflammasome assembly and activation of the NLRP3 inflammasome. Ghre significantly preserves the activities of antioxidant-related genes such as superoxide dismutase, catalase and malondialdehyde, retains concentrations of GSH and decreases levels of ROS [95]. During the progression of neurodegeneration or brain disorders, microglia are activated, intensifying neuron death [96]. However, whether microglia are primed first by some endogenous and exogenous factors or the activation of these resident immune cells precedes neuronal cell death remains to be explored. Apoptotic caspase 8 and 3/7 signaling is known to be a microglia activator [97]. Findings show that Ghre treatment suppresses activated microglia and reduces neuronal death and microglial activation [30]. Reducing the activation of caspase 3 and both the Bcl proapoptotic protein Bax and Bcl-2, Ghre prevents the apoptotic pathway activation at the mitochondrial level [98]. Therefore, Ghre mediates neuroprotective effects by reducing apoptosis and consequent inflammation through reduced microglial activation mediated by caspase exerting neuroprotection. The modulation of microglial cells could play a crucial role in basic hypothalamic functions such as the regulation of energy balance and in conditions of neurodegeneration, cognitive impairment and behavioral disorders [99].

## 9. Microglia and Ghre in AD

AD is the most common form of dementia. The pathological features of AD consist of amyloid-β (Aβ) settles, neurofibrillary tangles and neuronal injury [100]. In particular, the Aβ removal alteration leads to microglial activation, causing the release of proinflammatory cytokines, which in turn increase amyloid precursor protein (APP) production and consequently more Aβ formation [101]. AD frequently coincides with other comorbidities such as MetS, which includes central obesity, insulin resistance, atherogenic dyslipidemia and hypertension [102]. Insulin resistance and vascular endothelial dysfunction are the main causes of this association. Experimental animal models have indicated the link between MetS and cognitive dysfunction. Additionally, behavioral alterations and variations of inflammatory markers have been observed [103]. Mice fed a diet rich in advanced glycation end products develop MetS and cognitive dysfunctions [104]. Ghre guarantees adequate serum glucose throughout the fasting period. Several studies have focused on the interactions of Ghre with the insulin system in humans [105,106]. It has been proposed that antagonizing the Ghre insulin static system makes it possible to maintain glucose homeostasis in humans [107].

Ghre might improve cognition in AD via a CNS mechanism involving insulin signaling. The investigation using a murine model fed a high-glycemic-index diet treated with the Ghre agonist revealed a persistent challenge for glucose homeostasis in AD. The Ghre agonist impairs glucose tolerance immediately after administration but not in the long term. Immunoassay analysis showed a beneficial impact of long-term treatment on insulin signaling pathways in hippocampal tissue. Moreover, the Ghre agonist improved spatial learning in mice, raised their activity levels and reduced their body weight and fat mass [108]. It was previously shown that p-IRS-1 Ser636 is associated with peripheral insulin resistance and obesity. Mice treated with the Ghre agonist exhibit a reduced expression of p-IRS-1 Ser636 and AD [109,110].

These findings indicate the importance of Ghre agonists on cognitive effects in AD, particularly affecting hippocampal brain areas and functions. Ghre decreases peripheral glucose uptake in periods of fasting, whereas it improves or unalters uptake in CNS energy deficiency conditions [111].

On the other hand, in vivo neurodegeneration studies have shown how Ghre administration prevents the death of neuronal cells induced by kainic acid (KA), promoting astroglia and microglia inactivation by regulating the expression of COX-2, TNF-α and IL-1β in the hippocampal area. In particular, Ghre acts by blocking KA-induced MMP-3 expression in hippocampal neurons [112]. In another study conducted on endothelial cells, it was reported that Ghre enrichment blocked the production of MMP-3 in a GHSR-dependent pathway, providing a possible mechanism for Ghre-induced inhibition of microglia activation [113].

Taken together, all these data support the interplay between Ghre and microglia activation also through the inhibition of MMP-3 expression.

In AD, the main sources of cytokines, which contribute to neuroinflammation development, are microglia and astrocytes [114]. Scientific evidence from AD patients and AD animal models showed accumulations of microglia at sites of insoluble fibrillar Aβ protein (fAβ) deposition. fAβ triggers microglia through several receptors such as TLR, and CD36 is one of the main scavenger receptors involved in the internalization of oxidized low-density lipoprotein (LDL) [115,116]. The binding of fAβ to CD36 triggers a signaling cascade that controls microglial recruitment, activation and secretion of ROS and inflammatory mediators, leading to neuronal dysfunction and death [115]. Additionally, we have seen how only nonacylated Ghre is able to regulate the expression and production of cytokines, such as IL-6 and IL-1β, in microglial cells. It should be taken into account that the action of the peptide could also occur through modulation of the CD36 receptor, an alternative GH receptor, in the activation of Aβ-induced microglia. As previously speculated, given the absence of the receptor at the level of the microglial cell membrane, it is possible to deduce that this mechanism could act indirectly or through a GHSR-A1-independent pathway [117]. Nonacylated Ghre also binds the CD36 receptor by interfering with the activation of the Aβ peptide of CD36 in microglia, suggesting that, in AD, the protective role of Ghre may occur independently of GHSR-1A [118]. In humans and rodents, Ghre exhibits anti-inflammatory effects inhibiting inflammatory cytokine expression in monocytes and T cells, reducing signs of joint inflammation [27,119]. Treatment with Ghre or GHSR-1A agonists reduces Aβ peptide levels and inflammation of microglia [120]. The Aβ oligomers induce depolarization of the mitochondrial membrane, which is reduced by the action of Ghre that is also able to prevent the activation of glycogen synthase kinase-3 beta [121].

Once microglia become activated in response to neurodegenerative events, they are able to synthesize proteolytic enzymes such as cathepsin B. causing extracellular matrix damage and further neuronal dysfunction [122]. Overall, microglial and proinflammatory mediators might play a relevant role in the progression of AD. Ghre has no significant effects on IL-1β and IL-6 mRNA levels in fAβ-stimulated N9 cells [118]. Nevertheless, Ghre stimulates the proliferation of adult rat hippocampal progenitor cells. Cholinergic synapse loss might result from microglia release of ROS and other inflammatory neurotoxic mediators. Circulating Ghre enters the hippocampus and stimulates dendritic spine synapse formation and the generation of long-term potentiation phenomena in neurons of the hippocampal region [123]. The toxicity of fAβ is almost absent in cultures of brain cells from which microglia had been removed, indicating that microglia are needed to mediate the cytotoxic effects of fAβ on primary neurons [124]. In vitro experiments with desacyl-Ghre demonstrate that it is able to efficiently blunt the increase in IL-1β and IL-6 mRNA induced by fAβ [118]. AD is also connected to the hunger alterations responsible for body weight decrement. An anti-inflammatory diet can inhibit neuroinflammation associated with AD and attenuate neuroinflammation via indirect immune pathways from the gut microbiome and systemic circulation. Diet may influence cognitive aging via several inflammatory pathways [125]. Ghre may be a preventive strategy mediating the neuroprotective effect and improving cognitive skills [126]. Numerous in vitro and in vivo studies have been focused on how Ghre influences memory performance. In several AD animal model studies, it has been shown that Ghre plays a role in AD pathogenesis, contributing to Aβ, synaptic loss, neuroinflammation, and cognitive dysfunction. Moon et al. 2011 showed that Ghre rescues memory deficits by acting on AβO-induced microgliosis and attenuating hippocampal neuron death [127]. Additionally, Ghre modulates tau phosphorylation, which is another major cause of AD pathogenesis in hippocampal neurons; however, no clinical trials have reported positive results, and further studies are necessary [128]. AD is also associated with neuronal loss caused by apoptosis that contributes to AD. Ghre prevents ROS increase, calcium deregulation and mitochondrial dysfunction regulating mitochondria well-being [129]. Collectively, these findings led us to consider that Ghre may be a possible key for the treatment that affects AD progression (Table 1).

## 10. PD, Microglia and Ghre

PD is the second most common neurological disease characterized by the progressive degeneration of the nigrostriatal system producing a dopamine deficit [130]. The clinical manifestations of PD are bradykinesia, muscle stiffness and uncontainable tremor [131]. The development of PD is associated with the degeneration of DA neurons in the brain. These neurons easily undergo degeneration because of the energy request to send signals along their extensive branching. The formation of Lewy bodies containing aggregated alpha-synuclein (α-syn) is the histological feature of PD [132]. Epidemiological evidence suggests an association between PD and preexisting metabolic dysfunctions. Diabetes is a risk factor and a negative prognostic factor for PD [133]. In fact, PD patients show reduced striatal dopamine transporter (DAT) binding and a faster motor and cognitive decline in the presence of diabetes. Diabetic patients show signs of subclinical striatal DA dysfunction on DAT scans even in the absence of PD [134]. Inflammation is a common feature of both diabetes mellitus type 2 and PD [135]. Defective insulin signaling, increased oxidative stress, mitochondrial dysfunction and neuroinflammation are pathological mechanisms producing the exacerbation of PD [134]. An increase in the number of microglial cells has been observed in PD lesions [136]. Evidence suggests that Ghre secretion is altered in PD [137]. Moreover, altered Ghre signaling altered DA neurons, and its administration positively influenced microglial deactivation and mitochondrial metabolism [25,104]. In fact, in an in vivo model of PD study, Bayliss et al. showed that Acyl-Ghre exogenous administration protects against neurodegeneration, also increasing the spine synapse number and neurogenesis in the adult hippocampal neurons [138]. If on one side Ghre has no effect on insulin but is active at low plasma levels of glucose, on the other side, insulin is active at high levels of glucose and counteracts Ghre action. Conversely, Ghre activates a number of growth factor signaling pathways that can compensate for a loss of insulin signaling. Moreover, Ghre has impressive protective effects in animal models of PD. The administration of Ghre reduces the neurodegeneration of substantia nigra and the turnover of dopamine observed in the MPTP model of PD and also reduces the chronic inflammatory response in the brain. Ghre activates AMPK and lipid oxidation and improves mitochondrial activity and genesis [139].

It is known that GHSR-1A and DA receptor 2 (D2) interact. GHSR-1A dimerizes with several G-protein-coupled receptors, such as GHSR-1B, melanocortin 3 receptor (MC3), D1, D2 and serotonin 2C receptor (5-HT2C). Ghre and GHSR-1A binding trigger the heterodimer formation with D2, causing DA release and microglial activation [140].

Moreover, Ghre is known to act on peripheral macrophages inhibiting LPS-induced release of proinflammatory cytokines, and LPS leads to DA neuron death through microglia induction [141]. We can speculate that Ghre may also act on microglia inhibiting LPS induction on cytokine release, supporting a coaction of both Ghre and microglia in causing PD (Table 2).

## 11. Conclusions

In this review, we looked at the correlation between Ghre and microglia (Table 3).

Much evidence supports that Ghre behaves as a metabolic hormone, being involved in feeding behavior and in the regulation of energy homeostasis. In addition, contributing to the preservation and compliance of neuronal activity and connectivity, Ghre exerts a protective role on the CNS. It shares similar properties to those of neuroactive peptides and internal messengers in the brain. Therefore, inducing Ghre signaling in the CNS improves neuroplasticity, neuroprotection and cognitive functions, promoting endogenous repair mechanisms in the brain, thereby reducing the possibility of neurodegenerative diseases. A growing number of investigations suggest that counteracting Ghre signaling recovers glucose control.

In 2005, Dixit and Taub questioned whether Ghre is a hormone or a cytokine [142]. Ghre, like cytokines, is a powerful mediator; thus, it is easy to suppose a double function for this molecule. Currently, in the literature, the exact mechanisms of Ghre action on microglia have not yet been clearly outlined. Consequently, we can hypothesize the existence of a Ghre-signaling-dependent pathway by which damaged neurons could communicate with glia, which in turn can stimulate neighboring cells, including microglia (Figure 3). Overall, these observations suggest that further investigations are needed to better understand the mechanisms that allow Ghre to interact with microglial cells, in order to provide new targets for the treatment of neurodegenerative disorders involving metaflammation and cell excitotoxic death.

## Figures and Tables

**Figure 1 ijms-23-13432-f001:**
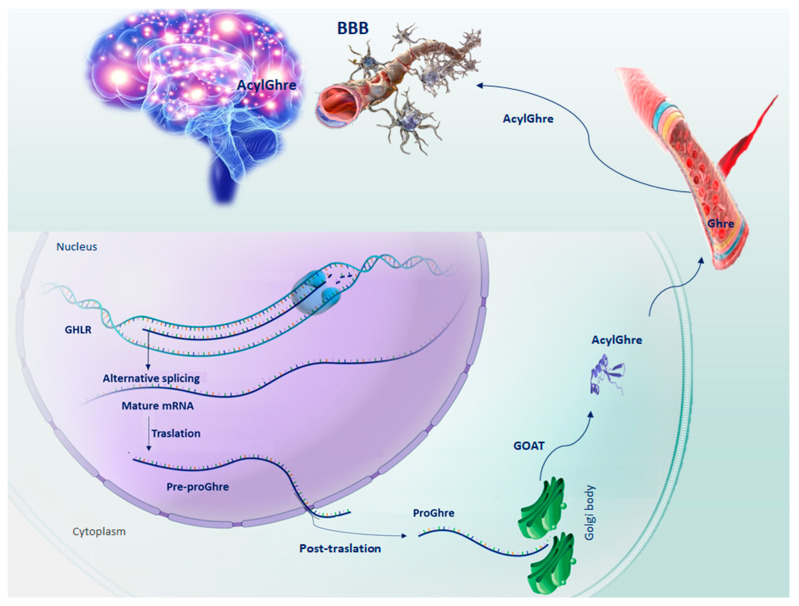
Schematic representation of the sequential steps of Ghrelin production. GHLR gene, located on the short arm of chromosome 3, through splicing alternative process, generates the mRNA from which the pre-proGhre precursor will originate. In turn, pre-proGhre is cleaved to pro-Ghre and transported to the Golgi body, where it is acylated by the action of the GOAT enzyme. The acylated form of Ghre through the circulatory system reaches and crosses the BBB, carrying out its functions in the brain. Abbreviation = AcylGhre, acylated Ghrelin; BBB, blood–brain barrier; Ghre, Ghrelin; GHLR, human ghrelin gene; GOAT, Ghre-O-acyltransferase; Pre-proGhre, pre-proGhrelin; ProGhre, pro-Ghrelin.

**Figure 2 ijms-23-13432-f002:**
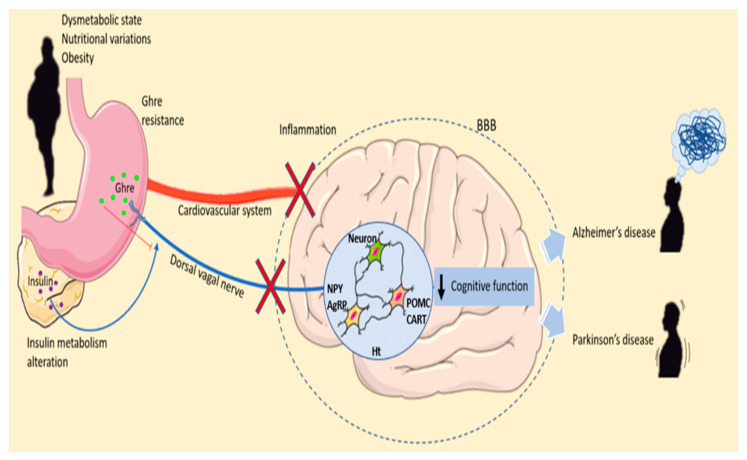
Ghrelin’s roles in malnutrition and neurodegenerative diseases. Dysmetabolic state, nutritional variations and obesity can lead to Ghre resistances and insulin metabolism alteration. Ghre is synthesized by the stomach and released into the peripheral blood. Moreover, Ghre reaches the brain both by crossing the and by the dorsal vagal nerve. In the Ht, Ghre stimulates the release of orexigenic neuropeptide stimulating appetite. Obesity alters insulin metabolism. In the fasting conditions, AgRP expression increases, whereas POMC expression is reduced. During the fed state, the level of AgRP is reduced, and POMC levels increase, stimulating energy expenditure and altering the metabolic feedback loop. Inflammation of the hypothalamus due to malnutrition and dysmetabolism can lead to cognitive function impairment promoting neurodegenerative disease development. Abbreviation = AgRP, agouti-related peptide; BBB; blood–brain barrier CART, cocaine- and amphetamine-regulated transcript; Ghre, Ghrelin; Ht, hypothalamus; NPY, neuropeptide Y; POMC, proopiomelanocortin; The figure was produced using Servier Medical Art.

**Figure 3 ijms-23-13432-f003:**
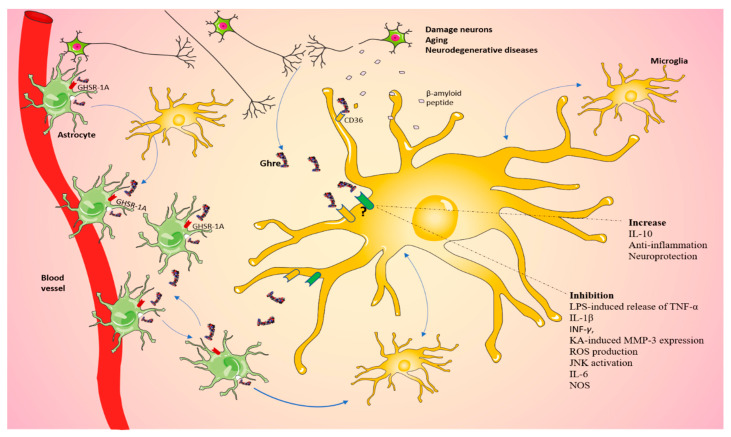
Ghrelin regulates microglia in the CNS. Schematic view of our proposed mechanism by which damaged neurons could communicate with glia through a Ghre-signaling-dependent pathway, which in turn can stimulate neighbor cells, including microglia. The question mark indicates that the role played by the pathway is still controversial. The figure was produced using Servier Medical Art.

**Table 1 ijms-23-13432-t001:** Effect of Ghrelin on Alzheimer disease (AD).

Study Type	Model	Effect	References
In vitro	N9 microglia cells	Desacyl-ghrelin, hexarelin and EP80317 are capable in vitro to blunt the increase in IL-1β and IL-6 mRNA induced by fAβ25–35 in N9 cells. The effects of fAβ25–35 on IL-1β mRNA levels were attenuated by desacyl-ghrelin, hexarelin and EP80317, but not ghrelin.	[118]
In vitro	Hippocampal neurons	Leptin and ghrelin protect hippocampal neurons from Aβ oligomer-induced toxicity. Leptin and ghrelin prevent oxidative stress and mitochondrial dysfunction. Leptin and ghrelin revert GSK3β activation induced by Aβ oligomers. Neuroprotection by leptin and ghrelin occurs in a receptor-mediated manner.	[121]
In vivo/In vitro	Ghrelin KO in a mouse model of AD	Ghrelin deletion affects memory performance, and acyl ghrelin treatment may delay the onset of early events of AD.	[126]
In vivo/In vitro	Male ICR mice	Systemic injection of ghrelin rescues cognitive impairment induced by AßO in vivo, suggesting that ghrelin-mediated restoration of behavioral performance on cognitive deficit is at least in part mediated by inhibition of microgliosis and impairment of neuronal integrity.	[127]
In vivo	Tg APPSwDI mouse	The ghrelin agonist impaired glucose tolerance immediately after administration, but not in the long term. The ghrelin agonist improved spatial learning in the mice, raised their activity levels and reduced their body weight and fat mass. Ghrelin might improve cognition in Alzheimer’s disease via a central nervous system mechanism involving insulin signaling	[108]
In vivo	Tg APPSwDI mouse	Treatment with a hunger-inducing ghrelin agonist is sufficient to reduce AD-related cognitive deficits and pathology in Tg AD model mice.	[120]

Abbreviations = AßO, amyloid-β oligomer; GSK-3β, glycogen synthase kinase-3 beta; IL-1β, interleukin 1 β; IL-6 mRNA, interleukin-6 messenger RNA. Mice strain: ICR, Institute of Cancer Research (ICR) mice; Tg APPSwDI: human APP with Swedish, Dutch, and Iowa mutations on a C57BL/6 background.

**Table 2 ijms-23-13432-t002:** Effect of Ghrelin on Parkinson’s disease (PD).

Study Type	Model	Effect	References
In vivo/ In vitro	C57Bl/6 mice	Administration of ghrelin significantly attenuated the loss of substantia nigra pars compacta neurons and the striatal dopaminergic fibers. Ghrelin reduced nitrotyrosine levels and improved the impairment of rotarod performance. In vitro administration of ghrelin prevented 1-methyl-4-phenylpyridinium-induced dopaminergic cell loss, MMP-3 expression, microglial activation and the subsequent release of TNF-α, IL-1β and nitrite in mesencephalic cultures.	[30]
In vivo	A53T and wild-type mice	Ghrelin administration in PD mice did not affect weight gain in wild-type mice but improved weight loss in PD mice. Attenuation of dopaminergic neuron loss in substantia nigra and a low level of dopamine content in the striatum occur in PD mice with ghrelin treatment.	[89]
In vivo	PD patients	In PD patients with weight loss, higher active ghrelin concentration in plasma was not observed nor was an increased appetite.	[137]
In vivo	GOAT KO and Ghre KO mice	Acylated Ghrelin is the isoform responsible for in vivo neuroprotection by attenuating dopamine cell loss and glial activation.	[138]

Abbreviations = IL-1β, interleukin 1 beta; MMP-3, metalloproteinase-3, stromelysin-1; PD, Parkinson’s disease; TNF-α, tumor necrosis factor-alpha. Mice strain: A53T mutants α-synuclein (α-syn) *are associated with Parkinson’s disease (PD)* and wild-type α-syn; C57Bl/6 mice are inbred; GOAT KO mice are acylated ghrelin octanoyl-acyltransferase (GOAT) knockout (KO) mice.

**Table 3 ijms-23-13432-t003:** Effect of Ghrelin on microglia.

Study Type	Effect	References
In vitro	In N9 microglia cells, which express the CD36 receptor, 10(–7) M desacyl-ghrelin prevents the stimulation effects of fAβ (25–35) on IL-6 mRNA levels. Similarly, on IL-1β, mRNA levels are reduced by desacyl-ghrelin form.	[118]
In vitro	In primary cultured hippocampal neurons, Ghre reduces amyloid-β oligomer production of superoxide and mitochondrial membrane depolarization. Moreover, it improves cell survival and inhibits cell death. Ghre prevents glycogen synthase kinase 3β activation.	[121]
In vivo	In primary hippocampal neurons, Ghre acts on M1 microglia/macrophages, improving neurological function and reducing cerebral infarction, apoptotic cells and IL-1β and TNF-α expression.	[65]
In vivo	Ghre blocks kainic acid-induced MMP-3 expression in hippocampal neurons, preventing neuronal cell death induced by kainic acid, promoting astroglia and microglia inactivation through COX-2, TNF-α and IL-1β regulation.	[112]
In vivo	Ghre agonist reduces AD pathology and improves cognition in AD mouse model, improves the performance in the water maze and reduces levels of amyloid beta and microglial activation	[120]
	Abbreviations = AD, Alzheimer’s disease; COX-2, cyclooxygenase-2; IL-1β: interleukin-1 beta; LPS: lipopolysaccharides; MMP-3: metalloproteinase-3, stromelysin-1; N9: cell line N9 inhibitors; TNF-α: tumor necrosis factor-alpha;	

## Data Availability

References for this review were identified through searches of Pubmed for articles published from 1991 to 2022.

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
