# Peer review of "The Interplay between Ghrelin and Microglia in Neuroinflammation: Implications for Obesity and Neurodegenerative Diseases"

_ijms, 2022, doi:10.3390/ijms232113432_

Round 1

Reviewer 1 Report

This review aims to evaluate Ghre crosstalk with microglial cells, trying to understand how it can accomplish its key anti-inflammatory role.

I found this paper interesting and well-written with an high scientific impact.

I have a minor point to address: the authors should better explain (in the end of the introduction section) what is known from the literature on the topic and what they want to show as new evidence to increase the added value and the originality of their review. This could be also inserted in the conclusions section.

Author Response

Dear reviewer,

First of all, we would like to thank for reviewing the paper and suggesting improvements.

Following your suggestion in the abstract we added your suggested sentence: “This review aims to summarizes what is known from the current literature on the way in which Ghre modulates microglial activity during neuroinflammation and their impact neurometabolic alterations in neurodegenerative diseases”

We hope you are satisfied with our answer

Reviewer 2 Report

The proposed review manuscript entitled „The interplay between Ghrelin and microglia in neuroinflammation: implications for obesity and neurodegenerative diseases“ by Russo et al. aims to comprehend the role of neuropeptide ghrelin in microglia homeostasis and neuroinflammation. The Abstract is very well written. However, the major part of the main text is complex and does not contain good organization and structure that would help to understand more clearly the function of ghrelin in these processes. For better structure/organization of the manuscrit please look at the review „The Effects of Leptin on Glial Cells in Neurological Diseases„ by Fujita and Yamashita in Front. Neurosci., 2019. Firstly, the authors need to describe more clearly the expression of ghrelin and its receptors, the structure, expression in different tissues etc. Figures that describe this are welcome. Overall, the complexity of the ghrelin role in neuroinflammation and neurodegeneration needs to be simplified and more clearly described. The authors might include additional subsections on distinct effects of ghrelin or may distinguish the obtained results in vitro from those in vivo. In the present form, the data are presented in a mixed form which makes the review hard to read and to follow. In addition, the Figures and Tables included do not substantially contribute to the paper nor are appropriately described. For example, in Figure 1 the cells that release NPY are not depicted/identified. In the figure legend it is clear nor is explained what happens to AgRP, POMC and CART under these conditions. Furthermore, in the whole manuscript negative effects of ghrelin application/treatment have not been addressed. The English requires additional editing.  

Author Response

Dear reviewer,

First of all, we would like to thank for reviewing the paper and suggesting improvements.

Please, find below a point-by point review of your valuable comments. We hope you are satisfied with our answers

  1. An extensive revision of the text has been carried out to improve the organization and structure in order to better understand the mechanisms by which ghrelin modulates microglia activity during obesity-induced neuroinflammation
  2. In the section 2 we have more clearly described the structure, the expression of ghrelin at its receptors and their function in various tissues. As you suggested has been added a new figure
  3. The reference “Fujita e Yamashita in Front. Neurosci., 2019” has been added
  4. In the section 3. Ghre role in neuroinflammation and neurometabolism, we have tried to describe in a clearer and simpler way the role Ghre in these processes
  5. We did not include additional subsections on the effects distinct in vitro from the in vivo effects of ghrelin because the suggested subdivision made the manuscript too fragmentary and difficult to read, so we preferred to keep the tables, which have been modified by better ordering in vitro and in vivo experiments.
  6. In the legend in Figure 1 we explained what happens to AGrp, POMC and CART
  7. English has been reviewed by a native language teacher

Round 2

Reviewer 2 Report

The revised version of the manuscript has been significantly improved. The authors have incorporated almost all corrections. Few minor point are still missing:

1. in Figure 1 the authors need to add a short description of the figure following the title. Also, few abbreviations are missing for Pre-proGhre, ProGhre, AcylGhre.

2. In Table 2 - Models used need to be described in more detailed. Add mouse strain or cell lines used for in vivo and in vitro studies, respectively. Also, an abbreviation for ICR mouse is missing.

3. In Table 3 - in the second study listed the effect of ghrelin was analyzed on primary hippocampal neurons not on microglia as depicted in the table's title.

Author Response

Dear reviewer,

First of all, we would like to thank for reviewing the paper and suggesting further improvements.

Following your suggestions;

In Figure 1:  the description of the figure and the missing abbreviations for Pre-proGhre, ProGhre, AcylGhre. have been integrated

A more detailed description of model used for the in vivo and in vitro studies has been added, in both Table 1 and 2, as you suggested.

Moreover, the abbreviation for ICR mouse has been added.

In Table 3, the expression “primary hippocampal neurons” has been added.